# Correlation between MRI Features of Adenomyosis and Clinical Presentation

**DOI:** 10.3390/diagnostics13172749

**Published:** 2023-08-24

**Authors:** Youn-Jee Chung, Sung Eun Rha, Mee-Ran Kim, Yu Ri Shin

**Affiliations:** 1Department of Obstetrics & Gynecology, Seoul St. Mary’s Hospital, College of Medicine, The Catholic University of Korea, Seoul 06591, Republic of Korea; porshe80@catholic.ac.kr (Y.-J.C.); mrkim@catholic.ac.kr (M.-R.K.); 2Department of Radiology, Seoul St. Mary’s Hospital, College of Medicine, The Catholic University of Korea, Seoul 06591, Republic of Korea; sungeun.rha@gmail.com

**Keywords:** adenomyosis, uterus, magnetic resonance imaging, classification, diagnosis

## Abstract

This study aimed to explore the correlation between MRI features, clinical risk factors, and symptoms associated with adenomyosis. Overall, 112 patients with pathologically confirmed adenomyosis were included in this retrospective study. MRI findings and clinical presentation, including visual analog scale (VAS) scores, cancer antigen 125 (CA-125) and hemoglobin levels, and parity, were analyzed. Additionally, 131 patients undergoing active surveillance were included to validate the MRI parameters and clinical presentations. Associations between MRI parameters and adenomyosis-related clinical presentations were assessed. Patients with operated adenomyosis were younger and had larger lesions, which were more frequently of the diffuse type and posterior localization, coexisting ovarian endometriosis, deep infiltrating endometriosis, myometrial cysts, and diffusion restriction than the non-operated lesions (*p* < 0.05). Patients with operated adenomyosis also exhibited higher VAS scores and CA-125 levels, and nulliparity was more common in this group (*p* < 0.05). In contrast, patients with non-operated adenomyosis showed a higher frequency of entire localization and fibroids (*p* < 0.05). Among the MRI parameters, size and classification were associated with the VAS and CA-125 levels. Myometrial cysts were associated with CA-125 levels. Classification was also associated with hemoglobin levels, and posterior localization was associated with parity. We identified a significant correlation between MRI features and clinical presentation in patients with adenomyosis.

## 1. Introduction

Adenomyosis is a prevalent gynecological disease characterized by the presence of endometrial glands and stroma within the myometrium and affects approximately 20% of women worldwide [1]. Traditionally, adenomyosis has been diagnosed using histopathology following hysterectomy in perimenopausal women experiencing abnormal uterine bleeding or pelvic pain. However, this condition is not as well-known as other gynecologic conditions, such as endometriosis, which introduces challenges in obtaining an accurate diagnosis. In addition, patients with adenomyosis may have future pregnancy plans or a desire to preserve their uteruses owing to recent trends of delayed marriage and pregnancy; therefore, many may choose uterine-preserving surgeries rather than hysterectomies. Recent advances in imaging techniques, including ultrasound (US) and magnetic resonance imaging (MRI), have enabled widespread noninvasive diagnosis of the condition. Adenomyosis has garnered increasing attention owing to its high prevalence on US and MRI and its variable clinical symptoms, including pelvic pain, abnormal uterine bleeding, and pregnancy-related disorders [2,3].

The spectrum of adenomyosis ranges from the thickening of the junctional zone to focal or diffuse lesions involving the entire uterine wall, resulting in diverse manifestations and complex diagnostic classifications. Furthermore, no consensus on shared clinical and imaging diagnostic criteria exists, and data from previous studies are heterogeneous and not fully comparable. Few studies have aimed to establish a correlation between the MRI phenotypes of adenomyosis and clinical outcomes [4,5,6,7]. However, most MRI studies lack satisfactory multiparametric imaging and validation, and the relationship between the imaging features of adenomyosis and symptoms or other adverse outcomes, such as infertility or pregnancy loss, remains unclear.

MRI is an accurate and objective imaging modality for obtaining anatomical information about adenomyosis. However, universal criteria for predicting disease severity are currently unavailable. Consequently, the clinical importance of MRI in assessing disease severity, predicting clinical risk factors, and evaluating symptomatology remains unknown. A comprehensive understanding of the relationship between imaging features and clinicopathological characteristics is crucial for determining disease severity and enabling tailored therapeutic approaches for each adenomyosis subtype. Therefore, our study aimed to investigate the correlation between the MRI features and clinical presentation of the disease, including risk factor profiles, signs, and symptoms.

## 2. Materials and Methods

### 2.1. Patients Population

The study was approved by the appropriate Institutional Review Board, and the requirement for informed consent was waived owing to the retrospective design. We consecutively investigated 112 patients with adenomyosis who underwent uterine-preserving surgery at our tertiary academic center from December 2011 to January 2020. Surgeries included conservative laparotomic uterine-sparing surgery (*n* = 36) and robot-assisted uterine-sparing surgery (*n* = 76). Indications for surgical treatment included progressive anemia, exacerbation of clinical symptoms that caused abdominal compression and discomfort in daily life, and severe pelvic pain that was difficult to control. Additionally, 131 patients with adenomyosis who underwent active surveillance and wanted to preserve their uterus, including management with hormone therapy, were included. Women with coexisting malignancies were excluded (Figure 1). All patients underwent routine pelvic MRI examinations, except during the menstrual phase. Demographic data were retrieved from the electronic medical records linked to centralized computer systems. Data on patient age, gravidity, parity, and adenomyosis-related findings, including dysmenorrhea, as measured by the visual analog score (VAS), initial hemoglobin level, and serum cancer antigen 125 (CA-125) levels, were collected.

### 2.2. MRI Examination and Image Analysis

All MRI examinations were performed using a 3.0-T scanner (Verio, Siemens Healthcare, Erlangen, Germany) with a pelvic phased-array coil. The patients fasted for 3 h and received an antispasmodic drug (Buscopan, Boehringer Ingelheim, Ingelheim, Germany) intravenously immediately before imaging to reduce bowel peristalsis. The MRI protocol included turbo spin-echo T2-weighted sequences acquired in the sagittal, axial, and coronal planes with radial blades (BLADE: repetition time/echo time (TR/TE), 4000 ms/118 ms; slice thickness, 6 mm; flip angle, 138°; matrix, 320 × 320; field of view (FOV), 240 × 240 mm). Axial T1-weighted sequences were acquired with and without fat suppression (TR/TE, 650 ms/12 ms; slice thickness, 6 mm; flip angle, 150°; matrix, 320 × 224; FOV, 240 × 240 mm). Diffusion-weighted imaging sequences were axial single-shot isotropic echo-planar sequences with fat saturation. Free-breathing diffusion-weighted imaging was acquired with b-values of 0, 50, and 1000 s/mm2 (TR/TE, 6200 ms/80 ms; slice thickness, 6 mm; flip angle, 90°; matrix, 100 × 100; FOV, 240 × 240 mm). An apparent diffusion coefficient map was automatically generated. Gadoteridol (ProHance, Bracco, Milan, Italy) at a dose of 0.2 mM/kg was injected intravenously at a rate of 1.0 mL/s, followed by a 20 mL saline flush. After contrast injection, sagittal and axial T1-weighted fat-suppressed gradient-echo images (TR/TE, 650 ms/12 ms; slice thickness, 6 mm; flip angle, 150°; matrix, 320 × 224; FOV, 240 × 240 mm) were obtained.

MRI images were retrospectively reviewed based on size and classification (internal, diffuse, external) parameters. Adenomyosis was classified based on the affected area and the degree of myometrial infiltration, according to the references [7,8]. Adenomyosis was classified into the following types according to the degree of infiltration: internal type, defined as that occurring in the inner uterine myometrium without affecting the outer structures of the myometrium; external type, defined as that arising in the outer myometrium, including adenomyomas, or that invading from outside the uterus, disrupting the serosa but not affecting the junctional zone; and diffuse type, defined as that being in an advanced stage that could not be classified as an internal or external type. Regarding the affected area, adenomyosis was classified according to localization (anterior, posterior, lateral, fundal, or entire), presence of a concomitant pathology (ovarian endometrioma (OMA), deep infiltrating endometriosis (DIE), or fibroid), margin (ill-defined or clearly demarcated), presence of myometrial cysts, presence of diffusion restriction, and degree of enhancement compared to the myometrium (less, equal, or more enhancement).

### 2.3. Statistical Analyses

Associations between MRI parameters and adenomyosis-related clinical presentations were assessed using the chi-squared, Fisher’s exact, Wilcoxon rank-sum, or Kruskal–Wallis tests, whenever appropriate. All statistical analyses were performed using SAS version 9.4 (SAS Institute, Cary, NC, USA). Statistical significance was set at *p* < 0.05. Pairwise deletion was applied to the missing data.

## 3. Results

The MRI findings and clinicodemographic characteristics of patients with operated adenomyosis (adenomyomectomy) (*n* = 112, 39.5 ± 5.2 years old) compared to those with non-operated adenomyosis (*n* = 131, 44.3 ± 7.1 years old) are summarized in Table 1. Age, size, classification, posterior and entire localization, coexisting OMA, DIE, and fibroid, myometrial cysts, diffusion restriction, enhancement degree, VAS scores, CA-125 levels, and parity showed statistically significant differences between the two groups. The operated group appeared to be younger and presented with larger lesions than the non-operated group. Posterior localization, presence of OMA, DIE, and myometrial cysts, and diffusion restriction were more frequently observed in the operated group than in the non-operated group. The proportions of internal, diffuse, and external adenomyosis were 4.5%, 53.6%, and 42%, respectively, in the operated group. In addition, the proportions of adenomyosis with less, equal, and more enhancement than the myometrium were 24.3%, 71.2%, and 4.5%, respectively, in the operated group. Regarding clinical parameters, the operated group exhibited higher VAS scores and CA-125 levels than the non-operated group. In addition, the operated group presented with a nulliparous state more frequently than the non-operated group. Entire localization and fibroids were more frequently observed in the non-operated group than in the operated group (Figure 2).

Table 2 shows the association between the MRI parameters and VAS scores. Size and classification were associated with VAS scores in both the non-operated and operated groups (*p* = 0.004, 0.001, and *p* = 0.007, 0.001, respectively). The VAS score was higher for lesions > 5 cm than those < 5 cm in both groups. The VAS scores tended to increase in the order of diffuse, external, and internal adenomyosis in both groups. The VAS scores appeared to be associated with posterior localization, coexisting fibroids, and myometrial cysts in the operated group (*p* = 0.008, 0.009, and 0.001, respectively). The VAS scores tended to be higher with posterior localization, no fibroids, and the presence of myometrial cysts.

Table 3 shows the association between the MRI parameters and CA-125 levels. Size, classification, and myometrial cysts were associated with CA-125 levels in both the non-operated and operated groups (*p* < 0.001, *p* < 0.001, *p* < 0.001 and *p* < 0.001, *p* < 0.001, *p* = 0.002, respectively). CA-125 levels were higher in lesions > 5 cm than those < 5 cm in both groups. CA-125 tended to increase in the order of diffuse, external, and internal adenomyosis in both groups.CA-125 levels were higher with the presence of myometrial cysts in both groups. CA-125 levels were associated with posterior localization and coexisting fibroids in the non-operated group (*p* = 0.041 and 0.003). CA-125 levels tended to be higher in patients with posterior localization and no fibroids.

Table 4 shows the association between the MRI parameters and hemoglobin levels. Classification was associated with hemoglobin levels in both the non-operated and operated groups (*p* = 0.017 and *p* = 0.029, respectively). In the non-operated group, hemoglobin levels tended to be lower in cases of diffuse adenomyosis than in those of external or internal adenomyosis. Hemoglobin levels tended to be lower in cases of internal adenomyosis compared to those of diffuse or external adenomyosis in the operated group. Hemoglobin levels appeared to be associated with myometrial cysts and diffusion restriction in the non-operated group (*p* = 0.004 and 0.019, respectively). Hemoglobin levels tended to be lower in the presence of myometrial cysts and diffusion restriction. Hemoglobin levels appeared to be associated with size and anterior localization in the operated group (*p* = 0.014 and 0.032, respectively). Hemoglobin levels tended to be lower with lesions > 5 cm and anterior localization.

Table 5 shows the association between the MRI parameters and parity. Posterior localization was associated with parity in both the non-operated and operated groups (*p* = 0.041 and *p* = 0.031, respectively). A nulliparous status tended to correlate with posterior localization in both groups. Classification, and coexisting OMA, DIE, and myometrial cysts, were associated with parity in the non-operated group (*p* = 0.027, 0.002, 0.04, and 0.04, respectively). Nulliparous patients presented more frequently with external adenomyosis, had a higher frequency of coexisting OMA and DIE, and presented with more myometrial cysts than multiparous patients. Fundal localization significantly correlated with parity in the operated group (*p* = 0.027). Multiparous patients presented more frequently with lesions with fundal localization than nulliparous patients.

## 4. Discussion

We aimed to identify the imaging findings associated with adenomyosis-related clinicopathological characteristics, including pain, tumor marker levels, bleeding, and parity. Notably, patients with operated adenomyosis exhibited several distinct characteristics compared to those with non-operated adenomyosis: they tended to be younger and have larger lesions, which were mostly of a diffuse type and with posterior localization, coexisting OMA, DIE, and myometrial cysts, and diffusion restriction. Additionally, patients who underwent surgery for adenomyosis had higher VAS scores and CA-125 levels, with nulliparity being more common in this group. Adenomyosis often coexists with other pelvic pathologies, including endometriosis and fibroids. Fibroids are present in approximately 37% of adenomyosis cases in women undergoing hysterectomy and may mask the presence of adenomyosis [9]. Adenomyosis accompanied by fibroids is commonly observed in the absence of surgery. In our study, the non-operated group had a higher prevalence of entire localization or fibroids, indicating a preference for medication-based treatments over surgery.

Our study found that lesion size and classification were associated with the VAS scores and CA-125 levels. The severity of symptoms, such as dysmenorrhea, is associated with the number or density of ectopic endometrial glands, mainly based on lesion size [10,11,12,13]. Our results support these findings and highlight the relationship between tumor size and CA-125 levels. Furthermore, the posterior localization of adenomyosis was more likely to be associated with a higher VAS score in the operated group. Previous studies have suggested that a thickened posterior myometrial layer is a risk factor for painful cramping during menstruation [14,15]. Myometrial cysts are highly characteristic of adenomyosis; however, they were only identified on MRI in approximately half of the cases [16]. Some studies have not observed a correlation between the presence of myometrial cysts and clinical manifestations [5,17,18]. However, we found a significant association between myometrial cysts and CA-125 levels. Additionally, we observed a clear association between myometrial cysts and higher VAS scores in the operated group, suggesting that the relative number of glands within the adenomyotic mass can affect the clinical symptoms and influence the choice of surgical treatment. Additionally, our results demonstrate that hemoglobin levels were associated with adenomyosis classification. The diffuse type was associated with lower hemoglobin levels than the external or internal type. These findings are consistent with those of previous studies that reported heavier menstrual bleeding in patients with diffuse adenomyosis [11,13,19]. However, other studies did not find a relationship between the depth of myometrial involvement and menorrhagia [10,20,21]. Patients with adenomyosis are believed to have lower pregnancy and implantation rates [22]. However, the specific adenomyosis phenotype resulting in infertility remains unclear. In our study, nulliparous women tended to present with adenomyosis in a posterior localization. Bourdon et al. noted that infertility was related to focal adenomyosis of the outer myometrium, but not to diffuse internal adenomyosis [5]. However, no statistically significant differences in pregnancy rates were identified between patients with localized and diffuse adenomyosis [22]. Limited evidence exists regarding the association between infertility and the adenomyosis phenotype.

Adenomyosis is a heterogeneous condition and comprises multiple subtypes with distinct imaging features. Recently, several subtype classifications based on MRI findings have been proposed. Clinically, classifications can assist in diagnosis and prognosis assessments and guide the selection of an appropriate management modality, which ranges from conservative approaches to surgical interventions. Classifications differ in terms of terminology and categories; nevertheless, adenomyosis is commonly classified into three types: internal/intrinsic, external/extrinsic, and diffuse [7,8,23,24]. Our study found that the diffuse type was associated with higher VAS scores, elevated CA-125 levels, lower hemoglobin levels, and a higher likelihood of requiring surgery. In clinical practice, diffuse adenomyosis is often associated with more severe menstrual symptoms than focal localized disease [25]. The pain becomes more intense and intralesional bleeding more widespread when adenomyotic lesions deeply infiltrate the myometrial layer [11,26,27]. Elevated CA-125 levels have also been identified as a characteristic feature of diffuse adenomyosis.

Formerly, the diagnosis of adenomyosis lacked importance since it was considered a benign condition without specific treatment. However, the lack of accurate diagnosis may have led to unnecessary and ineffective procedures. Previously, patients who had completed their family plans and had severe symptoms of adenomyosis underwent a hysterectomy. However, the incidence of adenomyosis in women of childbearing age has been increasing due to recent trends in late marriage and delayed pregnancy. Therefore, the need for uterus- and fertility-preserving surgery is increasing. Early diagnosis is currently considered crucial for effective adenomyosis management, particularly in women seeking fertility preservation and symptomatic control. Recent attempts have been made to develop classification systems that integrate imaging findings with clinical symptoms and associated treatments [28]. For instance, internal/intrinsic adenomyosis is associated with bleeding symptoms and is less likely to respond to progestin therapy, whereas external/extrinsic lesions are linked to endometriosis and dysmenorrhea, and may have a better response to such therapy. Furthermore, the adenomyosis phenotype depicted by MRI can guide the selection of appropriate surgical treatments, such as adenomyomectomies for adenomyomas and cytoreductive surgeries for diffuse/internal adenomyosis [29]. Early diagnosis can be achieved by increasing awareness of the disease and implementing accurate diagnostic pathways, enabling timely intervention and improving patient outcomes.

No MRI-based classification for adenomyosis has been validated to date. Conflicting results from previous studies, which are often attributed to small sample sizes, variations in patient populations, and different definitions, cause difficulty in interpreting the clinical significance of the disease’s patterns. Comparative studies investigating imaging features are necessary to predict specific symptoms, assess disease severity, and evaluate treatment efficacy for adenomyosis. The establishment of a definite, evidence-based link between the MRI phenotype of adenomyosis and clinical outcomes would benefit both patients and clinicians. The standardized phenotype categorization would facilitate meaningful comparisons of symptoms and treatment outcomes, thereby enabling more tailored therapeutic approaches. In our study, multiparametric MRI parameters showed weaker associations with the clinical presentation than anticipated. Despite these limitations, we believe that our study addresses the pressing need for comprehensive research on the correlation between the MRI features of adenomyosis and its clinical presentation. We aimed to enhance our understanding of this prevalent gynecologic disorder by elucidating the relationship between its imaging findings, clinical risk factors, and symptomatology. The outcomes of this study may address the current knowledge gap, provide valuable insights into the clinical significance of MRI findings, and pave the way for improved diagnostic and therapeutic strategies for patients with adenomyosis.

Diffusion-weighted imaging visualizes water diffusion in tissues, which aids in malignancy assessment due to higher cellularity in malignant tumors. Most adenomyosis exhibits a low signal on high-b diffusion-weighted imaging, consistent with non-neoplastic features, and no diffusion restriction. Previous studies have highlighted the enhanced diagnostic accuracy of diffusion-weighted imaging compared to conventional MRI [30]. Adenomyosis may have a lower signal than the junctional zone and myometrium, with varied contrast enhancement, limiting diagnostic utility. Nonetheless, understanding these parameters could drive research on the correlation between MRI findings, adenomyosis severity and clinical outcomes.

The strength of our study is the detailed analysis of the MRI data, allowing for the description of specific phenotypic patterns, such as diffusion restriction, enhancement, and myometrial cysts. However, several limitations may have impacted our results. First, our broad inclusion criteria resulted in a heterogeneous study population, challenging the application of MRI parameters to specific patient groups based on menopausal status, symptomatology, and the presence of comorbidities, such as fibroids or endometriosis. Second, our analysis focused solely on patients who presented with symptoms, potentially overlooking individuals with asymptomatic adenomyosis. Nonetheless, these inclusion criteria were deemed appropriate considering our aim of evaluating the relationship between MRI features and clinical presentation.

## 5. Conclusions

In conclusion, we identified the MRI findings associated with adenomyosis-related clinicopathological characteristics. Among the parameters, lesion size was associated with the VAS scores and CA-125 levels. Myometrial cysts were associated with CA-125 levels. Classification was associated with the VAS scores and CA-125 and hemoglobin levels. The posterior localization of adenomyosis was associated with parity. This precise understanding of the spectrum of MRI features in adenomyosis is of great importance for enabling an accurate diagnosis, and facilitating individualized and appropriate management strategies for affected patients.

## Figures and Tables

**Figure 1 diagnostics-13-02749-f001:**
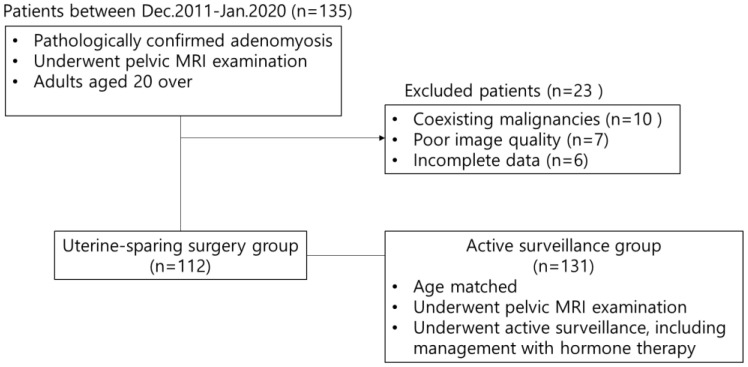
Flow chart of the study cohort.

**Figure 2 diagnostics-13-02749-f002:**
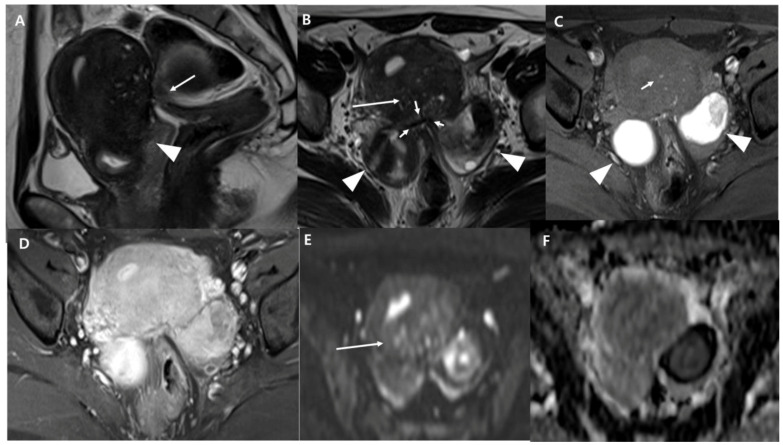
Operated adenomyosis in a 37-year-old nulliparous woman with severe dysmenorrhea (VAS 10) and CA-125 elevation. (**A**) Sagittal T2-weighted image shows an elevated posterior vaginal fornix (arrowhead) and tethered appearance of the rectum to the uterus (arrow), indicating Douglas obliteration. (**B**) Axial T2-weighted image shows a hypointense, ill-defined area (arrow) in the posterior myometrium and high-intensity cystic components within it. (**C**) Axial fat-suppressed T1-weighted images show high-signal-intensity spots, which correspond to some of the small high-intensity cystic components seen on the T2-weighted image. The hyperintense foci (short arrow) on T1W represent hemorrhage within the ectopic endometrial tissue. (**D**) The lesion shows an equal enhancement to myometrium on the contrast-enhanced T1-weighted image. (**E**,**F**) The mass shows (arrow) mild high-signal intensity on the diffusion-weighted image (**E**) acquired at b = 1000 s/mm^2^ and hypointensity on the apparent diffusion coefficient map (**F**). Bilateral endometriotic cysts are observed posterior to the uterus, appearing as a “kissing ovary” (arrowheads in (**B**,**C**)). At the posterior surface of the uterus, fibrotic irregular thickening is observed as a low signal intensity on T2WI (short arrows in (**B**)). The rectal wall is stretched strongly to the torus uterinus, suggesting severe adhesion. The bilateral ovaries and rectum converge at this point. Strong adhesion and strands were observed in the uterine serosal surface, rectum, and bilateral adnexa during the operation. The pouch of Douglas was closed by the adhesions. Surgical findings subsequently confirmed the presence of an adenomyoma with a bumpy and pinkish surface, which distinguishes it from a myoma.

**Table 1 diagnostics-13-02749-t001:** Comparison of clinical and MRI parameters between non-operated and operated groups.

	Total (*n* = 243)	Non–Op (*n* = 131)	Op (*n* = 112)	*p*-Value
Age (years) ^†^	42.1 ± 6.7 (38.0–47.0)	44.3 ± 7.1 (41.0–49.0)	39.5 ± 5.2 (36.0–43.0)	<0.001 *
**MRI parameters**				
Size (cm)	5.2 ± 2.7	4.6 ± 2.6	5.6 ± 2.7	0.003 *
	(3.1–7.0)	(2.5–6.0)	(3.5–7.2)	
Classification				<0.001 *
internal adenomyosis	43 (17.7)	38 (29.0)	5 (4.5)	
diffuse adenomyosis	108 (44.4)	48 (36.6)	60 (53.6)	
external adenomyosis	92 (37.9)	45 (34.4)	47 (42.0)	
Localization: anterior				0.962
No	208 (85.6)	112 (85.5)	96 (85.7)	
Yes	35 (14.4)	19 (14.5)	16 (14.3)	
Localization: posterior				<0.001 *
No	109 (44.9)	79 (60.3)	30 (26.8)	
Yes	134 (55.1)	52 (39.7)	82 (73.2)	
Localization: lateral				0.194
No	233 (95.9)	128 (97.7)	105 (93.8)	
Yes	10 (4.1)	3 (2.3)	7 (6.3)	
Localization: fundal				0.719
No	204 (84.0)	111 (84.7)	93 (83.0)	
Yes	39 (16.0)	20 (15.3)	19 (17.0)	
Localization: entire				<0.001 *
No	204 (84.0)	92 (70.2)	112 (100.0)	
Yes	39 (16.0)	39 (29.8)	0 (0.0)	
Concomitant ovary endometrioma				0.048*
No	157 (64.6)	92 (70.2)	65 (58.0)	
Yes	86 (35.4)	39 (29.8)	47 (42.0)	
Concomitant DIE				0.002 *
No	143 (58.8)	89 (67.9)	54 (48.2)	
Yes	100 (41.2)	42 (32.1)	58 (51.8)	
Concomitant fibroid				0.015 *
No	112 (46.1)	51 (38.9)	61 (54.5)	
Yes	131 (53.9)	80 (61.1)	51 (45.5)	
Margin				0.292
Ill-defined	220 (90.5)	121 (92.4)	99 (88.4)	
Clear demarcated	23 (9.5)	10 (7.6)	13 (11.6)	
Myometrial Cysts				<0.001 *
No	76 (31.3)	54 (41.2)	22 (19.6)	
Yes	167 (68.7)	77 (58.8)	90 (80.4)	
Diffusion restriction				0.019 *
Yes	46 (19.2)	18 (13.7)	28 (25.7)	
No	194 (80.8)	113 (86.3)	81 (74.3)	
Enhancement				0.01 *
Less than normal myometrium	49 (20.2)	22 (16.8)	27 (24.3)	
Equal enhancement to myometrium	188 (77.7)	109 (83.2)	79 (71.2)	
More enhancement than myometrium	5 (2.1)	0 (0.0)	5 (4.5)	
**Clinical Parameters**				
VAS Score ^†^	7.2 ± 2.7	6.5 ± 2.9	7.9 ± 2.3	<0.001 *
	(6.0–9.0)	(5.0–9.0)	(7.0–10.0)	
CA-125 (units/mL) ^†^	97.5 ± 199.3	56.1 ± 95.6	142.1 ± 262.8	<0.001 *
	(15.8–92.6)	(9.9–66.6)	(30.9–147.3)	
initial Hb (g/dL) ^†^	12.1 ± 1.8	12.3 ± 1.5	11.8 ± 2.0	0.18
	(11.1–13.4)	(11.4–13.3)	(10.4–13.4)	
Parity				<0.001 *
0	166 (68.6)	62 (47.7)	104 (92.9)	
1	29 (12.0)	23 (17.7)	6 (5.4)	
2	44 (18.2)	42 (32.3)	2 (1.8)	
3	3 (1.2)	3 (2.3)	0 (0.0)	

Unless otherwise indicated, data are the number of patients with the percentages in parentheses. ^†^ Values are means ± standard deviations, with ranges in parentheses. DIE = deep infiltrating endometriosis, VAS = visual analog scale, CA-125 = cancer antigen 125, Hb = hemoglobin. *p*-values were calculated using the chi-squared test, Fisher’s exact test, or Wilcoxon rank-sum test, wherever appropriate. Pairwise deletion was applied for missing data. * Values show statistically significant differences.

**Table 2 diagnostics-13-02749-t002:** Association between the MRI parameters and VAS scores.

	Non-Op (*n* = 116)			Op (*n* = 112)		
	*n*	Median (IQR)	*p*-Value	*n*	Median (IQR)	*p*-Value
Size			0.004 *			0.001 *
Size < 5, median	44	7.0 (3.0–8.0)		46	8.0 (6.0–9.0)	
Size ≥ 5, median	34	8.0 (7.0–9.0)		65	9.0 (8.0–10.0)	
Classification			0.007 *			0.001 *
Internal	27	6.0 (2.0–8.0)		5	5.0 (2.0–7.0)	
Diffuse	46	8.0 (6.0–9.0)		60	9.0 (8.0–10.0)	
External	43	7.0 (6.0–8.0)		47	8.0 (7.0–9.0)	
Localization: anterior			0.824			0.244
No	98	7.0 (5.0–9.0)		96	8.0 (7.0–10.0)	
Yes	18	8.0 (5.0–9.0)		16	9.0 (7.5–10.0)	
Localization: posterior			0.686			0.008 *
No	68	7.0 (3.0–9.0)		30	8.0 (6.0–9.0)	
Yes	48	8.0 (6.0–9.0)		82	9.0 (8.0–10.0)	
Localization: lateral			0.494			0.107
No	113	7.0 (5.0–9.0)		105	8.0 (7.0–10.0)	
Yes	3	8.0 (7.0–9.0)		7	7.0 (6.0–9.0)	
Localization: fundal			0.840			0.359
No	100	8.0 (5.0–9.0)		93	8.0 (7.0–10.0)	
Yes	16	7.0 (3.0–9.0)		19	8.0 (6.0–10.0)	
Localization: entire			0.755			
No	83	7.0 (5.0–9.0)		112	8.0 (7.0–10.0)	N/A
Yes	33	7.0 (3.0–9.0)		0	–	
Concomitant ovary endometrioma			0.550			0.32
No	80	8.0 (3.0–9.0)		65	8.0 (7.0–10.0)	
Yes	36	7.0 (6.0–8.0)		47	8.0 (7.0–10.0)	
Concomitant DIE			0.700			0.09
No	79	8.0 (3.0–9.0)		54	8.0 (7.0–9.0)	
Yes	37	7.0 (6.0–8.0)		58	8.0 (8.0–10.0)	
Concomitant fibroid			0.747			0.009 *
No	45	7.0 (6.0–9.0)		61	9.0 (8.0–10.0)	
Yes	71	7.0 (4.0–9.0)		51	8.0 (6.0–9.0)	
Margin			0.374			0.643
Ill-defined	107	8.0 (5.0–9.0)		99	8.0 (7.0–10.0)	
Clear demarcated	9	7.0 (5.0–8.0)		13	8.0 (8.0–9.0)	
Myometrial Cysts			0.096			0.001 *
No	43	7.0 (2.0–8.0)		22	7.0 (5.0–8.0)	
Yes	73	8.0 (6.0–9.0)		90	9.0 (8.0–10.0)	
Diffusion restriction			0.762			0.631
Yes	16	7.5 (4.5–8.0)		28	8.0 (7.5–10.0)	
No	100	7.0 (5.0–9.0)		81	8.0 (7.0–10.0)	
Enhancement			0.287			0.839
Less enhancement	17	8.0 (6.0–9.0)		27	8.0 (6.0–10.0)	
Equal enhancement	99	7.0 (4.0–9.0)		79	8.0 (7.0–10.0)	
More enhancement	0	–		5	8.0 (7.0–9.0)	

Data are VAS scores, with the ranges in parentheses. DIE = deep infiltrating endometriosis, N/A = not available, *p*-values were calculated using the Wilcoxon rank-sum test or Kruskal–Wallis test, wherever appropriate. * Values show statistically significant differences.

**Table 3 diagnostics-13-02749-t003:** Association between the MRI parameters and CA-125 levels.

	Non-Op (*n* = 117)			Op (*n* = 109)		
	*n*	Median (IQR)	*p*-Value	*n*	Median (IQR)	*p*-Value
Size			<0.001 *			<0.001 *
Size < 5, median	47	15.9 (9.8–35.0)		44	28.3 (14.9–64.6)	
Size ≥ 5, median	29	52.0 (23.9–83.6)		64	101.7 (56.0–200.3)	
Classification			<0.001 *			<0.001 *
Internal	32	9.7 (5.6–16.8)		4	23.3 (13.8–60.8)	
Diffuse	42	52.5 (18.0–108.4)		59	85.4 (50.8–177.9)	
External	43	27.2 (14.4–66.6)		46	41.1 (22.9–78.8)	
Localization: anterior			0.886			0.126
No	100	22.5 (10.1–65.9)		94	60.6 (28.0–126.9)	
Yes	17	18.0 (7.0–67.5)		15	93.7 (57.1–214.9)	
Localization: posterior			0.041 *			0.646
No	72	17.5 (7.4–56.5)		28	71.4 (23.4–128.2)	
Yes	45	28.0 (15.5–67.3)		81	66.1 (37.7–150.8)	
Localization: lateral			0.376			0.434
No	114	22.5 (10.3–67.3)		102	67.5 (33.6–147.3)	
Yes	3	9.8 (7.7–35.0)		7	27.1 (22.9–147.9)	
Localization: fundal			0.997			0.679
No	98	21.8 (9.9–66.6)		93	66.1 (30.9–147.3)	
Yes	19	25.5 (8.8–67.3)		16	82.7 (31.4–146.8)	
Localization: entire			0.215			
No	82	26.1 (14.4–66.6)		109	67.4 (30.9–147.3)	N/A
Yes	35	15.8 (7.2–76.3)		0	–	
Concomitant ovary endometrioma			0.096			0.318
No	79	20.4 (7.6–65.3)		63	58.1 (23.9–120.8)	
Yes	38	27.6 (15.5–67.3)		46	69.2 (37.9–150.8)	
Concomitant DIE			0.157			0.074
No	78	21.8 (7.6–67.3)		51	57.1 (22.9–100.7)	
Yes	39	23.2 (15.5–66.6)		58	73.3 (40.9–150.8)	
Concomitant fibroid			0.003 *			0.053
No	45	42.1 (18.8–70.3)		60	70.6 (42.0–171.3)	
Yes	72	16.0 (7.2–43.6)		49	55.0 (23.8–104.1)	
Margin			0.357			0.084
Ill-defined	108	23.6 (10.1–66.9)		96	70.3 (35.7–149.4)	
Clear demarcated	9	17.5 (7.7–20.4)		13	36.7 (13.6–58.1)	
Myometrial Cysts			<0.001 *			0.002 *
No	49	12.7 (6.8–22.4)		21	23.8 (11.4–55.9)	
Yes	68	39.6 (16.2–79.0)		88	71.8 (40.4–154.4)	
Diffusion restriction			0.684			0.878
Yes	16	27.2 (15.7–63.7)		28	66.8 (34.3–149.4)	
No	101	22.4 (9.8–66.6)		78	66.8 (28.7–147.3)	
Enhancement			0.354			0.934
Less enhancement	20	38.3 (19.3–65.2)		26	71.8 (27.1–120.8)	
Equal enhancement	97	20.4 (9.8–66.6)		77	66.1 (36.7–147.9)	
More enhancement	0	–		5	85.4 (21.8–150.8)	

Data are CA-125 levels (units/mL) with the ranges in parentheses. DIE = deep infiltrating endometriosis, N/A = not available. *p*-values were calculated using the Wilcoxon rank-sum test or Kruskal–Wallis test, wherever appropriate. * Values show statistically significant differences.

**Table 4 diagnostics-13-02749-t004:** Association between the MRI parameters and hemoglobin levels.

	Non–Op (*n* = 110)			Op (*n* = 112)		
	*n*	Median (IQR)	*p*-Value	*n*	Median (IQR)	*p*-Value
Size			0.071			0.014 *
Size < 5, median	39	12.8 (11.9–13.6)		46	13.0 (11.6–13.6)	
Size ≥ 5, median	32	12.3 (10.1–13.3)		65	11.5 (9.7–13.1)	
Classification			0.017 *			0.029 *
Internal adenomyosis	31	12.9 (12.5–13.9)		5	11.4 (8.7–12.1)	
Diffuse adenomyosis	43	12.0 (10.4–13.4)		60	11.6 (9.8–13.1)	
External adenomyosis	36	12.7 (11.8–13.3)		47	12.8 (11.3–13.6)	
Localization: anterior			>0.999			0.032 *
No	91	12.6 (11.5–13.3)		96	12.5 (11.1–13.4)	
Yes	19	12.9 (10.3–13.7)		16	10.4 (9.1–13.0)	
Localization: postesrior			0.558			0.911
No	71	12.7 (11.4–13.6)		30	12.2 (11.1–13.4)	
Yes	39	12.6 (11.5–13.3)		82	12.4 (10.0–13.4)	
Localization: lateral			0.569			0.487
No	108	12.6 (11.4–13.4)		105	12.2 (10.2–13.4)	
Yes	2	12.0 (11.3–12.7)		7	13.1 (11.1–13.5)	
Localization: fundal			0.539			0.16
No	93	12.6 (11.4–13.3)		93	12.6 (10.5–13.5)	
Yes	17	12.6 (11.5–13.9)		19	11.7 (9.5–13.0)	
Localization: entire			0.98			
No	75	12.6 (11.4–13.3)		112	12.2 (10.4–13.4)	N/A
Yes	35	12.6 (11.4–13.4)		0	–	
Concomitant ovary endometrioma			0.812			0.484
No	80	12.6 (11.4–13.6)		65	12.3 (10.1–13.3)	
Yes	30	12.8 (11.7–13.1)		47	11.9 (10.7–13.7)	
Concomitant DIE			0.224			0.622
No	77	12.7 (11.4–13.6)		54	12.6 (11.1–13.4)	
Yes	33	12.6 (11.5–12.9)		58	11.8 (10.0–13.4)	
Concomitant fibroid			0.136			0.058
No	41	12.4 (11.3–12.9)		61	12.6 (10.5–13.8)	
Yes	69	12.7 (11.7–13.4)		51	11.6 (9.8–13.1)	
Margin			0.781			0.073
Ill-defined	101	12.6 (11.4–13.3)		99	11.8 (10.0–13.4)	
Clear demarcated	9	12.6 (11.1–13.3)		13	12.8 (12.5–13.4)	
Myometrial cysts			0.004 *			0.284
No	43	12.9 (12.0–13.8)		22	12.9 (11.3–13.5)	
Yes	67	12.2 (11.1–13.1)		90	12.0 (10.1–13.3)	
Diffusion restriction			0.019 *			0.259
Yes	16	11.7 (10.0–12.7)		28	11.6 (10.2–13.2)	
No	94	12.7 (11.7–13.4)		81	12.5 (10.9–13.4)	
Enhancement			0.146			0.917
Less enhancement	18	12.8 (12.0–13.6)		27	12.3 (11.1–13.2)	
Equal enhancement	92	12.6 (11.3–13.3)		79	11.9 (10.0–13.4)	
More enhancement	0	-		5	11.7 (11.4–12.5)	

Data are hemoglobin levels (g/dL), with the ranges in parentheses. DIE = deep infiltrating endometriosis, N/A = not available. *p*-values were calculated using the Wilcoxon rank-sum test or Kruskal–Wallis test. * Values show statistically significant differences.

**Table 5 diagnostics-13-02749-t005:** Association between the MRI parameters and parity.

	Non-Op (*n* =130)			Op (*n* =112)		
	0 Nulliparous (*n* = 62)	1 or 2 or 3 Multiparous (*n* = 68)	*p*-Value	0 Nulliparous (*n* = 104)	1 or 2 or 3 Multiparous (*n* = 8)	*p*-Value
Size			0.139			0.064
Size < 5, median	22 (50.0)	27 (65.9)		40 (38.8)	6 (75.0)	
Size ≥ 5, median	22 (50.0)	14 (34.1)		63 (61.2)	2 (25.0)	
Classification			0.027 *			0.196
Internal adenomyosis	12 (19.4)	26 (38.2)		5 (4.8)	0 (0.0)	
Diffuse adenomyosis	23 (37.1)	25 (36.8)		58 (55.8)	2 (25.0)	
External adenomyosis	27 (43.5)	17 (25.0)		41 (39.4)	6 (75.0)	
Localization: anterior			0.305			0.599
No	55 (88.7)	56 (82.4)		88 (84.6)	8 (100.0)	
Yes	7 (11.3)	12 (17.6)		16 (15.4)	0 (0.0)	
Localization: posterior			0.041 *			0.031 *
No	32 (51.6)	47 (69.1)		25 (24.0)	5 (62.5)	
Yes	30 (48.4)	21 (30.9)		79 (76.0)	3 (37.5)	
Localization: lateral			0.605			0.414
No	60 (96.8)	67 (98.5)		98 (94.2)	7 (87.5)	
Yes	2 (3.2)	1 (1.5)		6 (5.8)	1 (12.5)	
Localization: fundal			0.793			0.027 *
No	53 (85.5)	57 (83.8)		89 (85.6)	4 (50.0)	
Yes	9 (14.5)	11 (16.2)		15 (14.4)	4 (50.0)	
Localization: entire			0.319			N/A
No	46 (74.2)	45 (66.2)		104 (100.0)	8 (100.0)	
Yes	16 (25.8)	23 (33.8)		-	-	
Concomitant ovary endometrioma			0.002 *			>0.999
No	36 (58.1)	56 (82.4)		60 (57.7)	5 (62.5)	
Yes	26 (41.9)	12 (17.6)		44 (42.3)	3 (37.5)	
Concomitant DIE			0.040 *			>0.999
No	37 (59.7)	52 (76.5)		50 (48.1)	4 (50.0)	
Yes	25 (40.3)	16 (23.5)		54 (51.9)	4 (50.0)	
Concomitant fibroid			0.336			0.726
No	27 (43.5)	24 (35.3)		56 (53.8)	5 (62.5)	
Yes	35 (56.5)	44 (64.7)		48 (46.2)	3 (37.5)	
Margin			0.747			0.233
Ill-defined	58 (93.5)	62 (91.2)		93 (89.4)	6 (75.0)	
Clear demarcated	4 (6.5)	6 (8.8)		11 (10.6)	2 (25.0)	
Myometrial cysts			0.040 *			0.654
No	20 (32.3)	34 (50.0)		20 (19.2)	2 (25.0)	
Yes	42 (67.7)	34 (50.0)		84 (80.8)	6 (75.0)	
Diffusion restriction			0.219			0.110
Yes	11 (17.7)	7 (10.3)		28 (27.7)	0 (0.0)	
No	51 (82.3)	61 (89.7)		73 (72.3)	8 (100.0)	
Enhancement			0.812			0.276
Less enhancement	11 (17.7)	11 (16.2)		23 (22.3)	4 (50.0)	
Equal enhancement	51 (82.3)	57 (83.8)		75 (72.8)	4 (50.0)	
More enhancement	-	-		5 (4.9)	0 (0.0)	

Data are the number of patients, with the percentages in parentheses. DIE = deep infiltrating endometriosis. *p*-values were calculated using the chi-squared test or Fisher’s exact test, wherever appropriate. * Values show statistically significant differences.

## Data Availability

The datasets generated and/or analyzed during the current study are available from the corresponding author upon reasonable request.

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
