# Peer review of "Correlation between MRI Features of Adenomyosis and Clinical Presentation"

_diagnostics, 2023, doi:10.3390/diagnostics13172749_

Round 1
Reviewer 1 Report
The authors analyzed MRI features from 243 patients and explored the correlation with clinical parameters (parity, sign, and symptom), retrospectively. It is worth to correlate MRI features with clinical factors.
The design and the presentation need to be improved.
- The majority of surgery for symptomatic adenomyosis is 'hysterectomy', not uterine-sparing surgery. This study deals with uterine-sparing surgery (USS) group and active surveillance (AS) group. This study doesn't include hysterectomy gorup. So, all the results and description should be narrowed down to 'uterine-sparing surgery.'
- It would be better to present the data from 'hysterectomy' group in the same period in your institution.
- I wonder why the authors presented data from USS group and AS group, separately in Table 2-5. I think there is no added value comparing with presenting data together from both groups.
- The authors should be clarify the study design. Is it case-control study? 'AS group' is described as 'control group' (Fig 1) and 'validating group.' I still wonder why the authors divide two groups when they present the data. All the patients checked MRI. All the data were not from surgery, but from MRI /clinical signs and symptoms/parity.
Please describe the reasons in detail.
- It would be great if you explain two MRI parameters (diffusion restrict, enhancement) more for gynecologists. What they mean in adenomysis?
- In Fig 2, there should be explanation on asterisk.
- I noticed there are missing data in Tables. Please describe these in the text.
- Both parity and tumor marker are not 'symptom.'
Author Response
Response to Reviewer 1 comments.
The authors analyzed MRI features from 243 patients and explored the correlation with clinical parameters (parity, sign, and symptom), retrospectively. It is worth to correlate MRI features with clinical factors.
The design and the presentation need to be improved.
- The majority of surgery for symptomatic adenomyosis is 'hysterectomy', not uterine-sparing surgery. This study deals with uterine-sparing surgery (USS) group and active surveillance (AS) group. This study doesn't include hysterectomy gorup. So, all the results and description should be narrowed down to 'uterine-sparing surgery.'
Response 1: Thank you for the comment. As you pointed out, our study results were limited to uterine-preserving treatment options for managing adenomyosis, such as uterine-preserving surgery or medical treatment, including hormonal therapy. We have described these points in the “Introduction”, “Materials and Methods” and “Discussion” section:
In addition, patients with adenomyosis may have future pregnancy plans or a desire to preserve their uteruses owing to recent trends of delayed marriage and pregnancy; therefore, many may choose uterine-preserving surgeries rather than hysterectomies.
The study was approved by the appropriate Institutional Review Board, and the requirement for informed consent was waived owing to the retrospective design. We consecutively investigated 112 patients with adenomyosis who underwent uterine-preserving surgery at our tertiary academic center between December 2011 and January 2020. Surgeries included conservative laparotomic uterine-sparing surgery (n= 36) and robot-assisted uterine-sparing surgery (n= 76). The indications for surgical treatment included progressive anemia, exacerbation of clinical symptoms that caused abdominal compression and discomfort in daily life, and severe pelvic pain that was difficult to control. Additionally, 131 patients with adenomyosis who underwent active surveillance and wanted to preserve their uterus, including management with hormone therapy, were included.
Previously, patients who had completed their family plans and had severe symptoms of adenomyosis underwent a hysterectomy. However, the incidence of adenomyosis in women of childbearing age has been increasing due to recent trends in late marriage and delayed pregnancy. Therefore, the need for uterus- and fertility-preserving surgery is increasing.
- It would be better to present the data from 'hysterectomy' group in the same period in your institution.
Response 2: Our study was limited to uterine-preserving treatments. Please consider the following points:
- I wonder why the authors presented data from USS group and AS group, separately in Table 2-5. I think there is no added value comparing with presenting data together from both groups.
Response 3: The purpose of this study was to identify the correlation between MRI features and clinical presentations in patients who underwent surgery for adenomyosis, and the USS group consisted of 112 patients. To confirm whether the MRI and clinical findings with characteristic associations in this USS group were characteristic associations only in the USS group, the AS group was additionally recruited. The USS and AS groups were analyzed separately to confirm the associations and identify variables that were not statistically significant in the AS group but were statistically significant in the USS group. Thus, our study aimed to evaluate MRI parameters to identify appropriate treatment modalities, such as uterine-preserving surgery. Therefore, we compared the USS and AS groups in the present study.
- The authors should be clarify the study design. Is it case-control study? 'AS group' is described as 'control group' (Fig 1) and 'validating group.' I still wonder why the authors divide two groups when they present the data. All the patients checked MRI. All the data were not from surgery, but from MRI /clinical signs and symptoms/parity.
Response 4: Thank you for this insightful comment. It was a cross-sectional study and not a case-control study. First, 112 USS groups were created. To identify the differences in MRI and clinical findings between the USS and AS groups, an additional AS group was formed by age-matching the USS group subjects approximately 1:1. The USS and AS groups were analyzed separately.
In Figure 1, we mistakenly described the AS group as the control group. We changed it to the AS group and modified the patient group to the USS group.
We have deleted the related phrase “to validate the MRI parameters and clinical presentation” from the text.
Please describe the reasons in detail.
- It would be great if you explain two MRI parameters (diffusion restrict, enhancement) more for gynecologists. What they mean in adenomysis?
Response 5: Thank you for the suggestion. In line with this comment, we have added the following sentence about two MRI parameters (diffusion restriction and enhancement) to the “Discussion” section:
Diffusion-weighted imaging visualizes water diffusion in tissues, which aids in malignancy assessment due to higher cellularity in malignant tumors. Most adenomyoses exhibit a low signal on high-b diffusion-weighted imaging, consistent with non-neoplastic features, and no diffusion restriction. Previous studies have highlighted the enhanced diagnostic accuracy of diffusion-weighted imaging compared to conventional MRI. (Quantitative diffusion-weighted magnetic resonance imaging of the normal and diseased uterine zones. Acta radiol. 2009;50:340-347) Adenomyosis may have a lower signal than the junctional zone and myometrium, with varied contrast enhancement, limiting diagnostic utility. Nonetheless, understanding these parameters could drive research on the correlation between MRI findings and adenomyosis severity and clinical outcomes.
- In Fig 2, there should be explanation on asterisk.
Response 6: In accordance with this comment, we have replaced the asterisk with multiple arrows to make it easier to understand and have added the following sentence to the figure legend:
The rectal wall is stretched strongly to the torus uterinus, suggesting severe adhesions.
- I noticed there are missing data in Tables. Please describe these in the text.
Response 7: Thank you for the suggestion. In line with this comment, we have added the following sentence to explain the missing data in the “Materials and Methods” section:
Pairwise deletion was applied to the missing data.
- Both parity and tumor marker are not 'symptom.'
Response 8: Thank you for your insightful comment. We have changed the ‘symptom’ into the ‘clinicopathological characteristics’ in the text.

Reviewer 2 Report
The main limitation is the heterogeneity of the study group in terms of age, symptomatology, desire to achieve pregnancy, etc.
The study presents a significant number of cases that were evaluated for adenomyosis by nuclear magnetic resonance imaging (MRI). Some of the patients were admitted and operated intraoperative examination and pathological anatomy confirmed the diagnosis obtained by MRI.
One hundred and thirty-one patients were examined and the data were statistically processed, giving the study power of evaluation even though the data obtained sometimes differed from the data reported in the literature.
The value of the study lies in the attempt to correlate MRI images for uterine adenomyosis with clinical signs and symptoms in deciding whether or not to operate on the cases presented, given that no scoring and evaluation systems for uterine adenomyosis are available in the literature.
The main limitation, as stated by the authors, is the heterogeneity of the study group in terms of age, symptomatology, desire to achieve pregnancy, etc.
Author Response
Response to Reviewer 2 comments.
- The main limitation is the heterogeneity of the study group in terms of age, symptomatology, desire to achieve pregnancy, etc.
Response 1: Thank you for your insightful comments. As pointed out by the reviewer, our broad inclusion criteria resulted in a heterogeneous study population. Unfortunately, we did not perform a subgroup analysis due to insufficient information during the data investigation. However, the findings of our study may be extrapolated to broader populations, thereby enhancing the generalizability of the study, including diverse groups to enhance its external validity. In response to this comment, we plan to conduct future research on specific subgroups to provide more targeted insights. By analyzing the outcomes within the subgroups, we can provide a clear understanding of how variations in the study population may have affected the overall findings.
- Comments on the Quality of English Language
Response 2: We have once again edited and proofread the English language and resubmitted our article.
